# Sum of Skinfold-Corrected Girths Correlates with Resting Energy Expenditure: Development of the NRG_CO_ Equation

**DOI:** 10.3390/nu16183121

**Published:** 2024-09-15

**Authors:** Diego A. Restrepo-Botero, Camilo A. Rincón-Yepes, Katherine Franco-Hoyos, Alejandra Agudelo-Martínez, Luis A. Cardozo, Leidy T. Duque-Zuluaga, Jorge M. Vélez-Gutiérrez, Andrés Rojas-Jaramillo, Jorge L. Petro, Richard B. Kreider, Roberto Cannataro, Diego A. Bonilla

**Affiliations:** 1Research Division, Dynamical Business & Science Society—DBSS International SAS, Bogotá 110311, Colombia; restrepo.diego@uces.edu.co (D.A.R.-B.); rincon.camilo@uces.edu.co (C.A.R.-Y.); lduque@dbss.pro (L.T.D.-Z.); jorge.velez@arthros.com.co (J.M.V.-G.); andres.rojasj@udea.edu.co (A.R.-J.); jlpetros@dbss.pro (J.L.P.); r.cannataro@gmail.com (R.C.); 2Grupo de Investigación NUTRAL, Facultad Ciencias de la Nutrición y los Alimentos, Universidad CES, Medellín 050021, Colombia; kfranco@ces.edu.co (K.F.-H.); magudelo@ces.edu.co (A.A.-M.); 3Research and Measurement Group in Sports Training (IMED), Faculty of Health Sciences and Sports, Fundación Universitaria del Área Andina, Bogotá 111221, Colombia; lcardozo11@areandina.edu.co; 4ARTHROS Centro de Fisioterapia y Ejercicio, Medellín 050012, Colombia; 5Research Group of Sciences Applied to Physical Activity and Sport, Universidad de Antioquia, Medellín 050010, Colombia; 6Research Group in Physical Activity, Sports and Health Sciences (GICAFS), Universidad de Córdoba, Monteria 230002, Colombia; 7Exercise & Sport Nutrition Laboratory, Human Clinical Research Facility, Texas A&M University, College Station, TX 77843, USA; rbkreider@tamu.edu; 8Department of Pharmacy, Health and Nutritional Sciences, University of Calabria, 87036 Rende, Italy; 9Hologenomiks Research Group, Department of Genetics, Physical Anthropology and Animal Physiology, University of the Basque Country (UPV/EHU), 48940 Leioa, Spain

**Keywords:** energy expenditure, indirect calorimetry, physical activity, validation study

## Abstract

Our study aimed to validate existing equations and develop the new NRG_CO_ equation to estimate resting energy expenditure (REE) in the Colombian population with moderate-to-high physical activity levels. Upon satisfying the inclusion criteria, a total of 86 (43F, 43M) healthy adults (mean [SD]: 27.5 [7.7] years; 67.0 [13.8] kg) were evaluated for anthropometric variables and REE by indirect calorimetry using wearable gas analyzers (COSMED K4 and K5). Significant positive correlations with REE were found for body mass (*r* = 0.65), body mass-to-waist (*r* = 0.58), arm flexed and tensed girth (*r* = 0.66), corrected thigh girth (*r* = 0.56), corrected calf girth (*r* = 0.61), and sum of breadths (∑3D, *r* = 0.59). As a novelty, this is the first time a significant correlation between REE and the sum of corrected girths (∑3CG, *r* = 0.63) is reported. Although existing equations such as Harris–Benedict (*r* = 0.63), Mifflin–St. Jeor (*r* = 0.67), and WHO (*r* = 0.64) showed moderate-to-high correlations with REE, the Bland-Altman analysis revealed significant bias (*p* < 0.05), indicating that these equations may not be valid for the Colombian population. Thus, participants were randomly distributed into either the equation development group (EDG, *n* = 71) or the validation group (VG, *n* = 15). A new model was created using body mass, sum of skinfolds (∑8S), corrected thigh, corrected calf, and age as predictors (*r* = 0.755, R^2^ = 0.570, RMSE = 268.41 kcal). The new NRG_CO_ equation to estimate REE (kcal) is: 386.256 + (24.309 × BM) − (2.402 × ∑8S) − (21.346 × Corrected Thigh) + (38.629 × Corrected Calf) − (7.417 × Age). Additionally, a simpler model was identified through Bayesian analysis, including only body mass and ∑8S (*r* = 0.724, R^2^ = 0.525, RMSE = 282.16 kcal). Although external validation is needed, our validation resulted in a moderate correlation and concordance (bias = 91.5 kcal) between measured and estimated REE values using the new NRG_CO_ equation.

## 1. Introduction

Total daily energy expenditure (TDEE) refers to the energy consumed by the body over a 24-h period. TDEE comprises the basal metabolic rate (BMR, or resting energy expenditure [REE], depending on how it is measured), the thermic effect of food (TEF), and the energy expenditure from physical activity, including both exercise and non-exercise activities [1,2]. BMR/REE is the energy an individual needs for vital physiological functions while at rest and represents 60–70% of TDEE [3]. TEF, on the other hand, is the metabolic energy required for digestion, absorption, and transport of food, accounting for 6–10% of TDEE [4]. Finally, the energy expenditure of physical activity refers to any musculoskeletal movement resulting in energy expenditure [5] and represents between 15–30% of TDEE [6]. Besides the energy costs of sustaining the human body and regular activities, the energy costs of stress-induced allostatic states (e.g., injury, disease, etc.), currently known as “allostasis and stress-induced energy expenditure” (ASEE), cannot be omitted [7].

Given its significant contribution to TDEE, various equations have been developed to estimate BMR/REE in different populations using predictor variables (e.g., physiological, anthropometric, etc.) [8,9,10,11,12,13]. Although popular equations such as Harris–Benedict, Schofield, Müller, Mifflin–St. Jeor, Cunningham, and others have advantages in terms of practicality and low cost [14,15], using formulas in populations other than those used in their development may overestimate or underestimate BMR/REE, as has been reported in obese women [16], older adults, and cancer patients [17]. Therefore, in the athletic population, the selection of a BMR/REE equation should be considered suitable only if it aligns with the demographics, physical characteristics, and sport of the athlete. Unfortunately, the lack of accuracy in estimating REE and energy intake could affect nutritional prescription and adherence to physical exercise programs, thereby hindering the achievement of goals and resulting in a lack of confidence in the process. It is important to highlight that the inadequate selection of a BMR/REE equation could increase the likelihood of suffering from Relative Energy Deficiency Syndrome, which raises the risk of musculoskeletal injuries, irritability, depression, decreased coordination, concentration, muscle strength, and cardiovascular endurance, among other effects [18].

To date, no equation has been developed to estimate REE in any Colombian population. Thus, the aim of this study is to develop alternatives for estimating REE in Colombian men and women with moderate-to-high physical activity levels. Using indirect calorimetry as a reference method, we aim to conduct the external validation of the equations commonly used in clinical practice: Harris–Benedict, Mifflin–St Jeor, and the “Food and Agriculture Organization of the United Nations/World Health Organization/United Nations” (WHO) equations. In addition, a novel equation, called NRG_CO,_ will also be developed. We hypothesize that the new NRG_CO_ equation will allow for the estimation of REE based on simple anthropometric variables associated with musculoskeletal mass, sex, and/or basic measurements (body mass and stature).

## 2. Materials and Methods

### 2.1. Study Design

The study was cross-sectional and designed based on international recommendations for multicentric research. It is reported according to the Strengthening the Reporting of Observational Studies in Epidemiology–Nutritional Epidemiology (STROBE–nut) guidelines, an extension of the STROBE statement [19,20].

### 2.2. Setting

This investigation was undertaken as part of the NRG Project of DBSS International SAS (NRG_DBSS, https://ichgcp.net/clinical-trials-registry/NCT05832710 (accessed on 31 May 2024)), with the support of Universidad CES, ARTHROS IPS, and Fundación Universitaria del Área Andina. Anthropometric and indirect calorimetry data were collected during the end of the second semester of 2023 and the first semester of 2024, as one of the thesis activities stipulated by the Master of Science in Sports Nutrition program at Universidad CES.

### 2.3. Participants

Men and women residing in Medellín and Bogotá or within the metropolitan area with moderate-to-high physical activity levels were recruited. The inclusion criteria were: (i) adults; (ii) individuals classified as having moderate or high physical activity levels according to the previously described IPAQ questionnaire; (iii) individuals who voluntarily signed the informed consent; (iv) individuals residing in the cities of Medellín or Bogotá and nearby municipalities in the metropolitan areas. Individuals over 60 years old, pregnant women, individuals diagnosed with cardiometabolic or respiratory diseases, or individuals with any type of musculoskeletal injury were excluded.

### 2.4. Variables

Primary outcomes of this study include basic measures, skinfolds (mm), girths (cm), and breadths (mm), according to the restricted profile established by the International Society for the Advancement of Kinanthropometry (ISAK). In addition, REE (kcal), assessed by indirect calorimetry, was used as a reference. Other anthropometric indices were calculated, and REE estimates were made using previously developed equations.

### 2.5. Data Sources/Measurement

The measurements of eligible participants were conducted at the facilities of the participating universities (Universidad CES and Fundación Universitaria del Área Andina) and research centers (ARTHROS IPS and DBSS) located in Medellín and Bogotá. To minimize technical errors during the measurements, assessments were performed between 8:00 and 17:00 (GMT-5) under controlled environmental conditions (<24 °C and <60% humidity). Additionally, standardized procedures by the DBSS Research Division were followed to develop and validate equations in different populations [21,22].

#### 2.5.1. Anthropometry-Based Analysis of Body Composition

Anthropometric measurements were carried out following the International Standards for Anthropometric Assessment established by ISAK [23]. Body mass was recorded using a digital scale with accuracy to the nearest 100 g (Seca 874, Hamburg, Germany). Stature was measured with a portable stadiometer with 1 mm graduations (Seca 213, Hamburg, Germany). The skinfold thickness of the triceps, subscapular, biceps, iliac crest, supraspinale, abdominal, thigh, and calf was measured with a calibrated skinfold caliper with constant closing compression of 10 g/mm^2^ (Harpenden, Baty Int., UK; Slim Guide, Creative Health, Ann Arbor, MI, USA). The girth measurements were carried out using a non-extensible metal Lufkin w606PM tape of 0.7 mm thickness (Apex Tool Group, Sparks, MD, USA). Breadth measurements were performed with small sliding calipers (Campbell 10, Rosscraft Srl, Buenos Aires, Argentina; Cescorf Equipamentos para Esportes, Porto Alegre, Brazil). Measurement error relative to technical standards by ISAK L2-certified anthropometrists was within acceptable limits (<5% for skinfolds, 1% for other measurements).

The sum of six and eight skinfolds (∑6S and ∑8S, respectively) were computed as absolute variables (expressed in millimeters), associated with whole-body adiposity as they correlate with whole-body fat mass [24]. The muscle girths for the arm, thigh, and calf were adjusted for skinfold thickness using the formula: girth − (π × skinfold) as a musculoskeletal index [25]. The sum of arm, thigh, and calf corrected girths (∑3CG) was computed, as well as the sum of humerus, bi-styloid, and femur breadths (∑3D). Other anthropometric indices evaluated as potential regressors include body mass-to-waist (BM/W) and waist-to-stature (W/Stature).

#### 2.5.2. Resting Energy Expenditure

To measure REE, we used the portable gas analyzer unit COSMED K4 (COSMED Srl, Rome, Italy) and the 4th generation wearable metabolic system COSMED K5 (COSMED Srl, Rome, Italy). Both devices have been validated previously and exhibit moderate-to-strong reliability [26,27,28,29]. For the assessment of REE, the recommendations established by the American Dietetic Association were followed to improve robustness and reproducibility [30].

Additionally, aware of the technical differences between basal metabolic rate and REE, we evaluated the validity of equations widely utilized in clinical practice: Harris–Benedict [31], Mifflin–St. Jeor [12], and the “Food and Agriculture Organization of the United Nations/World Health Organization/United Nations” (WHO) equation [32].

#### 2.5.3. Physical Activity Level

The participants were classified into three groups according to their level of physical activity (sedentary, physically active, and amateur athletes), using the validated short-form IPAQ questionnaire to identify the level of physical activity [33,34]. For this phase of the NRG Project, we only analyzed data from individuals with moderate-to-high levels of physical activity. The internal consistency of the instrument population was calculated using Cronbach’s alpha (in our population, α = 0.619 and 0.823 for moderate and vigorous activity, respectively).

### 2.6. Study Size

Based on previous recommendations [35], the a priori calculation of the sample size resulted in 90 participants for an R^2^ = 0.50 and a potentially accurate estimation outcome with three independent variables as regressors. In this study, a total of ninety physically active male and female Colombians (27.4 [7.7] years; 67.2 [13.6] kg; 167.2 [8.5] cm; 23.8 [3.6] kg/m^2^) were considered potentially eligible.

### 2.7. Statistical Methods

Descriptive statistics were reported as mean and standard deviation (SD), unless stated otherwise. To validate existing equations externally, we conducted correlation and agreement analyses between actual and estimated REE. This involved calculating the correlation coefficient (CC, as Pearson’s *r*), the coefficient of determination (R^2^), the adjusted coefficient of determination (aR^2^), and the root mean squared error (RMSE). Bland-Altman plots were utilized for concordance analysis.

Following the latest guidelines for enhancing data analysis practices and analytical methods from the DBSS Research Division [22,36], we employed the Yuen-Dixon test [37] utilizing 20% trimmed means (μ_t_) and 20% winsorized standard deviations (σ_w_) as a robust statistical approach for comparing samples of different sizes (i.e., EDG [*n* = 71] versus VG [*n* = 15]). This robust statistical method offers better control of Type I error when variances are unequal [38]. To develop the new equation, all possible combinations of independent variables in EDG were tested using the Ordinary Least Squares (OLS) approach with stepwise forward-backward variable selection. Zellner-Siow prior distributions on the regression coefficients were employed to compare all Bayes factors against the null model, and all potential models were ranked by their probability from highest to lowest. The Bayesian Adaptive Sampling (BAS) R package was used for this ranking process. The model’s explanatory power was assessed using the aR^2^. The SEE was calculated for all models to gauge regression precision, while the RMSE was used to measure the closeness of the estimated values to the actual values measured by DXA, as detailed elsewhere [21]. Additionally, all potential regression models were ranked based on the Akaike information criterion (AIC), the Bayesian information criterion (BIC), Mallows’ C_p_, and Hocking’s Sp. Following verification of all assumptions of multiple regression analysis (normality of residuals confirmed by the Omnibus k-squared and Jarque-Bera tests), the model with the best performance was chosen for further analysis. The predictability of the selected model was evaluated in the VG by computing the CC, R^2^, aR^2^, and RMSE. The agreement analysis for the new equation, NRG_CO,_ was conducted using Bland-Altman plots, with concordance intervals reported at 95% (limits of agreement, LoA). Statistical analyses were performed using the most recent version of the R statistical computing environment [39].

## 3. Results

### 3.1. Participants

After the call and assessment of 90 men and women who voluntarily participated in this study, four individuals were excluded due to inconsistent values. Therefore, a total of 86 apparently healthy adults met the inclusion criteria and were successfully evaluated for each of the primary variables in this study. Most participants belonged to the mestizo-white community, except for one participant who was Afro-descendant. The COSMED K5 device was used to evaluate 51 participants, while the COSMED K4 analyzer was used for the remaining 35 participants. Table 1 shows the characteristics of the participants.

A correlation analysis was performed to explore the relationship between the study variables and REE in all participants (Figure 1). Body mass (*r* = 0.65, 95% CI [0.51, 0.76]), BM/W (*r* = 0.58, 95% CI [0.42, 0.70]), arm flexed and tensed girth (*r* = 0.66, 95% CI [0.52, 0.77]), corrected thigh girth (*r* = 0.56, 95% CI [0.40, 0.69]), and corrected calf girth (*r* = 0.61, 95% CI [0.46, 0.73]) had a significant positive correlation with REE (*p* < 0.01). It is important to note that ∑3CG and ∑3D showed significant positive correlations with REE (*r* = 0.63, 95% CI [0.49, 0.75] and *r* = 0.59, 95% CI [0.43, 0.71], respectively). Full statistical results are reported in Appendix A.

### 3.2. External Validation of the REE Equations in the Colombian Population

All participants in this study were included in the external validation analysis of common equations to estimate REE. Moderate-to-high positive correlations were found for the Harris–Benedict (*r* = 0.631 [95% CI: 0.484, 0.744]), Mifflin–St. Jeor (*r* = 0.672 [95% CI: 0.537, 0.774]), and WHO (*r* = 0.641 [95% CI: 0.497, 0.751]) equations. It is worth noting that men showed higher correlation coefficient values than women (significant when comparing the 95% confidence intervals), as well as slightly non-significant superior correlation values for data collected in Medellín and those with high levels of physical activity (Figure 2).

Despite the above, none of the equations showed good agreement with the experimental measurements of REE. Bias for the estimated values of REE with Harris–Benedict (503.1 kcal [95% CI: 415, 591]; 95% LoA: −303, 1310), Mifflin–St. Jeor (296.3 kcal [95% CI: 230, 363]; 95% LoA: −310, 903), and WHO (224.0 kcal [95% CI: 156, 292]; 95% LoA: −401, 849) equations were all statistically significant (*p* < 0.05) for the Bland-Altman test (Figure 3). Overall, this suggests that common equations to estimate REE might not be valid in the Colombian population.

### 3.3. Development of the NRG_CO_ Equation

To create a new equation based on anthropometry for estimating REE in the Colombian population, participants (*n* = 86) were randomly allocated to either the equation development group (EDG, *n* = 71 [80%]) or the validation group (VG, *n* = 15 [20%]). There were no significant differences between the two groups (Table 1).

The candidate predictor variables considered for developing the new model using the OLS method included: age, sex, body mass, BM/W, flexed and tensed arm girth, corrected thigh girth, corrected leg girth, ∑3CG, ∑3D, and ∑8S. Following the assessment of all potential models using various combinations of predictor variables to estimate REE in kilocalories (a total of 1023 models expressed as Ŷ = β_0_ + β_1_X_1_ + β_2_X_2_ + β_3_X_3_ … + β_10_X_10_), the model demonstrating the best performance incorporated body mass, ∑8S, corrected thigh, corrected calf, and age as explainable variables. Although we assessed all potential combinations for estimating REE in datasets segmented by sex, the chosen model surpassed all other options following the model specification process (*r* = 0.755, R^2^ = 0.570 [0.37, 0.65], aR^2^ = 0.537, RMSE = 268.41 kcal). Table 2 shows the performance metrics of the selected model to estimate REE, while Appendix A contains all possible models, including all regressors.

An evaluation of the model’s assumptions was carried out using normality tests (Skewness and Kurtosis), linearity (Link Function), and homoscedasticity (Heteroscedasticity). All tests yielded *p*-values > 0.05, indicating that no significant deviations from the model’s assumptions were found. The normality of residual errors was verified using the Omnibus K-squared test (*p* = 0.254) and the Jarque–Bera test (*p* = 0.287), which assess skewness and kurtosis, respectively. Therefore, it is considered that the model’s assumptions are met, and the results of the analysis are valid.

The new NRG_CO_ equation (SEE = 280.52 kcal) to estimate REE in Colombian men and women with moderate-to-high physical activity levels is shown as follows: GER (kcal) = 386.256 + (24.309 × BM [kg]) − (2.402 × ∑8S [mm]) − (21.346 × Corrected Thigh [cm]) + (38.629 × Corrected Calf [cm]) − (7.417 × Age [years]).

Our additional Bayesian method for selecting the optimal regression model employed Zellner–Siow prior distributions for the regression coefficients to evaluate all Bayes factors against the null model, with potential predictors ranked by their probability from highest to lowest (Figure 4). This analysis allowed us to identify an alternative model with lower performance but simpler to estimate REE, including only body mass and ∑8S (*r* = 0.724, R^2^ = 0.525, aR^2^ = 0.511, RMSE = 282.16 kcal). This simpler equation is reported as the *fast* NRG_CO_ equation (SEE = 288.32 kcal) as follows: GER (kcal) = 641.482 + (21.433 * BM [kg]) − (2.702 × ∑8S [mm]).

### 3.4. Validation of the NRG_CO_ Equation

The validation process of the NRG_CO_ equation was performed with the VG sample (*n* = 15). It resulted in a non-significant moderate correlation (r = 0.482 [95% CI: −0.04, 0.797]). There was also moderate concordance (bias = 91.5 kcal; 95% LoA: −690, 873) between measured REE through indirect calorimetry and the estimated values using the NRG_CO_ equation (Figure 5).

## 4. Discussion

This study aimed to validate existing and commonly used equations to estimate BMR/REE in Colombia (Harris–Benedict, Mifflin–St. Jeor, and WHO). Our results are consistent with other studies in various populations [40,41,42], demonstrating that estimation equations can be inaccurate due to differences in population characteristics. In our study, the low *p*-values of the Bland-Altman test during external validation of the Harris–Benedict, Mifflin–St. Jeor, and WHO equations indicate strong evidence of significant bias suggesting that the methods are not interchangeable. This underscores that the existing equations are not adequate for accurately estimating REE in the physically active Colombian population. As a novelty, this is the first time that the sum of three corrected girths (∑3CG = arm + thigh + calf) and the sum of three breadths (∑3D: humerus + bi-styloid + femur) have been reported and evaluated for developing a model that estimates REE. Importantly, we found that ∑3CG showed a significant positive correlation with REE (*r* = 0.63, 95% CI [0.49, 0.75]) as well as the ∑3D (*r* = 0.59, 95% CI [0.43, 0.71]).

Considering the above, and given that no equation has been created for Colombians, we developed a new model to estimate REE with good performance metrics (NRG_CO_) using body mass, ∑8S, corrected thigh, corrected calf, and age as regressors (*r* = 0.755, R^2^ = 0.570 [0.37, 0.65], aR^2^ = 0.537, RMSE = 268.41 kcal). It should be noted that even though the coefficient of determination (R^2^) was 0.57 and might be considered a moderate fit, 43% of the variability is not accounted for by the model. However, several equations to estimate REE with R^2^ < 0.5 have been successfully developed in different populations, as reported in the recent systematic review by Ocagli et al. (2021) [43]. For example, in Mexican adults with excess adiposity, Orozco-Ruiz and co-workers (2017) developed a new equation to estimate REE with similar regression metrics to those reported in our study (*r* = 0.72, R^2^ = 0.51) [44]. In addition, several models developed by Mifflin–St. Jeor [12] have lower values than the R^2^ of our new NRG_CO_ equation. It is noteworthy to mention that non-linear approaches (e.g., machine learning algorithms) to estimate REE in acute kidney injury patients have been developed with performance metrics inferior to our model (*r* = 0.69, R^2^ = 0.48).

Our internal validation was performed in a small sample of participants (*n* = 15). The Bland-Altman statistics showed weak evidence for a statistically significant bias between the two methods (*p* value = 0.4), suggesting moderate agreement. It must be noted that bias for the estimated values of REE with the Harris–Benedict (503.1 kcal), Mifflin–St. Jeor (296.3 kcal), and WHO (224.0 kcal) equations were all statistically significant (*p* < 0.05) for the Bland-Altman test and considerably higher compared to the bias of the new NRG_CO_ equation (91.5 kcal). Also, the ICC during the internal validation was 0.45, indicating that measured and estimated REE have moderate agreement.

In the early 1990s, Spurr and colleagues reported a considerable intrasubject variation of basal metabolic rate (8.3%) in Colombian women [45]. Similarly, important variation has been reported across age groups in Colombian overweight females when aiming to estimate REE with regression equations [46]. This potential variability and hormonal fluctuations deserve further research. On the other hand, a large portion of the sample in our study exhibited a high level of physical activity (~83%), which would require adjustments in the model in future works, as we found subjects with very high resting energy expenditure (>2000 kcal). Although not measured, this would indicate socioeconomic differences in the participants, given previous reports have shown a relationship between physical activity level and the Colombian population [47]. Considering that TDEE is primarily composed of BMR/REE, which can account for up to 70% [11], nutrition and health professionals typically estimate REE first and then add values for TEF and physical activity expenditure using factors. It should be noted that allostatic load and stress could increase REE by approximately 9% to 67% in humans, depending on the intensity and duration of the stressor (e.g., mental stress, injury, etc.) [7]. Nevertheless, practitioners are encouraged to analyze that the ratio of REE to fat-free mass varies depending on body size and fat-free mass composition. For instance, individuals with lower body mass have a higher proportion of “residual mass” (metabolically less active tissue) and a lower proportion of metabolically active tissues like muscle and bone within their fat-free mass [48]. This highlights the importance of the new NRG_CO_ equation as it considers body composition when estimating REE.

Although there are various metabolic devices with rigorous validation and reliability studies [49,50,51], including the devices used in this study (COSMED K4 and K5), some authors suggest that portable devices may have lower reliability [52,53,54]. We acknowledge that adjusted models with better performance can be generated with a larger sample size, segmenting the analysis by city, utilizing the same metabolic device during data collection, or applying more advanced statistical techniques (machine learning algorithms). Nonetheless, all these limitations and ideas are being considered within the framework of the NRG Project, and we expect to report new or revised equations as more data is collected.

## 5. Conclusions

We demonstrated that equations commonly used in clinical practice to estimate REE (Harris–Benedict, Mifflin–St. Jeor, and WHO) might not be valid for physically active Colombian men and women. This study developed for the first time, a new equation to estimate REE in a Colombian population with moderate-to-high physical activity levels (*r* = 0.755, R^2^ = 0.570, RMSE = 268.41 kcal). This equation, called NRG_CO_, incorporates body mass, ∑8S, corrected thigh and calf girths, and age: REE (kcal) = 386.256 + (24.309 × BM) − (2.402 × ∑8S) − (21.346 × Corrected Thigh) + (38.629 × Corrected Calf) − (7.417 × Age). This study is the first to report a significant positive correlation between ∑3CG and REE, which underscores its applicability in field settings. Our Bayesian approach also allowed the development of a fast NRG_CO_ equation (*r* = 0.724, R^2^ = 0.525, RMSE = 282.16 kcal), which offers a simpler and faster version using only body mass and ∑8S: REE (kcal) = 641.482 + (21.433 × BM) − (2.702 × ∑8S). Given the small group of participants (*n* = 15) used for the internal validation, we invite the scientific community to perform the external validation of these new models. Overall, considering the significant contribution of REE to TDEE, these equations provide potentially more accurate REE estimates compared to existing methods for this specific population. The results of this study might help practitioners achieve greater accuracy in the assessment, design, and monitoring of nutritional and physical exercise interventions for aesthetic, health, or physical performance purposes.

## Figures and Tables

**Figure 1 nutrients-16-03121-f001:**
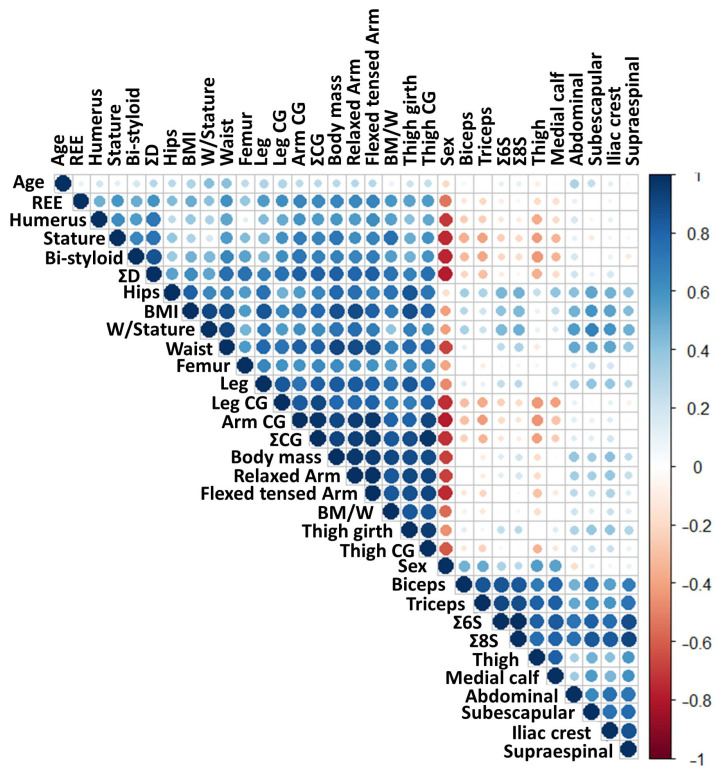
Draftsman correlation plot. Positive correlations are shown in blue, while negative correlations are depicted in red. The intensity of the colors and the size of the circles are proportional to the correlation coefficients. ∑CG: sum of corrected girths (arm + thigh + calf); ∑D: sum of breadths (humerus + bi-styloid + femur); ∑6S: sum of six skinfolds; ∑8S: sum of eight skinfolds; BM/W: body mass-to-waist ratio; CG: corrected girth; REE: resting energy expenditure; W/Stature: waist-to-stature ratio.

**Figure 2 nutrients-16-03121-f002:**
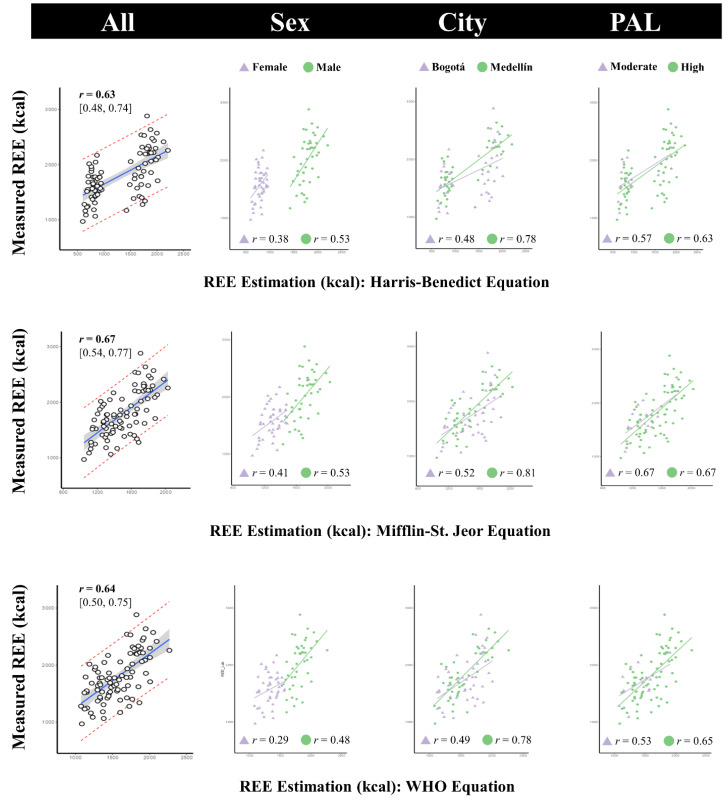
Correlation plots between measured REE (indirect calorimetry) and estimated REE with different equations. The figure also displays the correlation coefficients (Pearson’s *r*) by sex, city, and physical activity level (PAL).

**Figure 3 nutrients-16-03121-f003:**
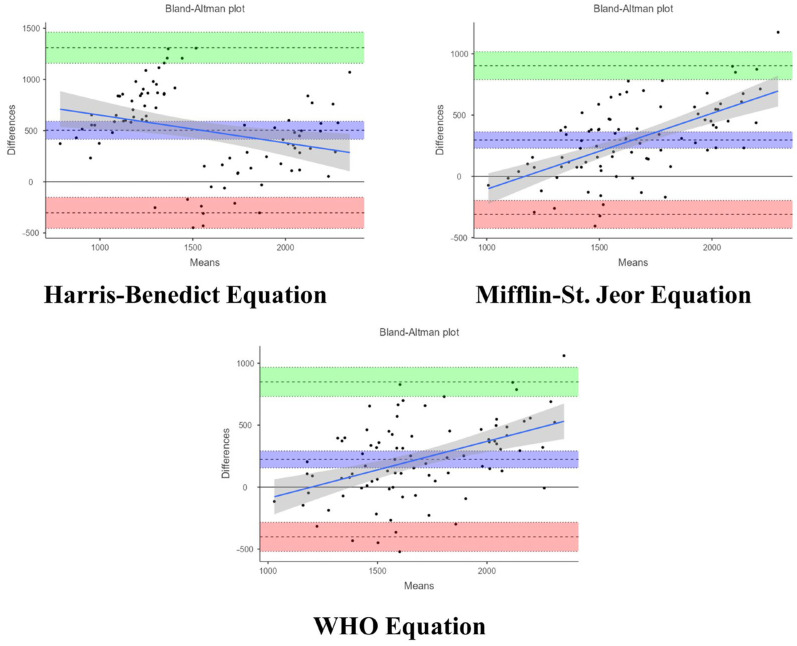
Bland–Altman plot showing the differences between measured and estimated REE using various equations. The plot displays individual differences between actual and estimated REE values against the average of the measured and estimated REE values. Bias (blue), upper (green), and lower (red) limits of agreement with their corresponding confidence intervals as well as the regression fit of the differences on the means (as solid blue line) are shown.

**Figure 4 nutrients-16-03121-f004:**
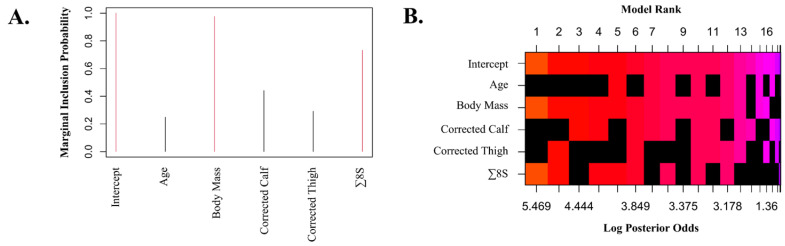
Bayesian selection of predictors. (**A**) Marginal posterior inclusion probabilities for each covariate are displayed, with probabilities exceeding 0.5 highlighted in red. (**B**) Zellner–Siow prior distributions for the regression coefficients are presented. Each row represents a variable or the intercept (indicated on the y-axis), while the x-axis represents the various models. Models are arranged from highest to lowest posterior probability, with the ranking shown on the top x-axis (each column corresponds to a different model). Variables not included in a model are depicted in black, while included variables are colored according to their log posterior probability, with orange indicating the model with the highest probability.

**Figure 5 nutrients-16-03121-f005:**
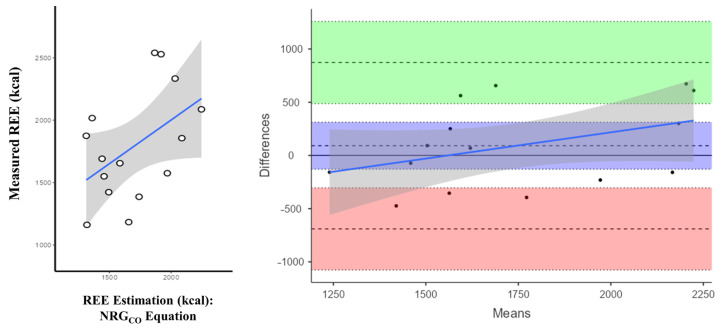
Correlation and concordance analysis of the NRG_CO_ equation. Correlation plot (**left**) and Bland–Altman plot (**right**) for differences between measured and estimated REE in kilocalories with the developed equation (NRG_CO_). Individual differences between real and estimated values are plotted against the mean of the values of measured and estimated REE. Bias (blue), upper (green), and lower (red) limits of agreement with their corresponding confidence intervals as well as the regression fit of the differences on the means (as solid blue line) are shown.

**Table 1 nutrients-16-03121-t001:** Characteristics of the study population.

Variable		All (*n* = 86) $\bar{x}$ (SD) [95% CI]	EDG (*n* = 71) $\bar{x}$ (SD) [95% CI]	VG (*n* = 15) $\bar{x}$ (SD) [95% CI]	ξ	*p* Value
Sex	Women	43 (50.00%)	35 (49.29%)	8 (53.33%)		
	Men	43 (50.00%)	36 (50.70%)	7 (46.66%)		
Region	Medellín	44 (51.16%)	38 (53.52%)	6 (40.0%)		
	Bogotá	42 (48.83%)	33 (46.47%)	9 (60.0%)		
PAL	Moderate	14 (16.27%)	12 (16.90%)	2 (13.33%)		
	High	72 (83.72%)	59 (83.09%)	13 (86.66%)		
Age (years)		27.5 (7.72) [25.8, 29.1]	27.4 (7.53) [25.6, 29.1]	28.1 (8.84) [23.2, 33.0]	0.137	0.738
Body mass (kg)		67.0 (13.8) [64.0, 69.9]	67.5 (13.9) [64.2, 70.9]	64.3 (13.1) [57.0, 71.5]	0.140	0.535
Stature (cm)		167.0 (8.65) [165, 169]	168.0 (8.52) [166, 170]	165.0 (9.13) [160, 170]	0.261	0.309
BMI (kg/m^2^)		23.8 (3.65) [23.0, 24.5]	23.8 (3.68) [22.9, 24.7]	23.5 (3.61) [21.5, 25.5]	0.127	0.985
Waist (cm)		76.1 (9.36) [74.1, 78.1]	76.0 (9.54) [73.8, 78.3]	76.2 (8.75) [71.4, 81.0]	0.133	0.976
BM/W (m/m)		87.3 (9.49) [85.3, 89.3]	88.1 (9.41) [85.8, 90.3]	83.6 (9.31) [78.5, 88.8]	0.429	0.091
W/Stature (cm/cm)		0.45 (0.04) [0.44, 0.46]	0.45 (0.04) [0.44, 0.46]	0.46 (0.04) [0.43, 0.48]	0.164	0.455
∑6S (mm)		84.5 (30.9) [77.9, 91.1]	84.3 (30.5) [77.1, 91.5]	85.2 (34.0) [66.4, 104]	0.149	0.758
∑8S (mm)		108.0 (39.4) [99.4, 116]	108.0 (38.6) [98.9, 117]	107.0 (44.2) [82.4, 131]	0.122	0.600
Arm CG (cm)		26.6 (5.10) [25.5, 27.7]	26.8 (5.25) [25.5, 28.0]	25.5 (4.28) [23.2, 28.0]	0.143	0.679
Thigh CG (cm)		48.2 (6.67) [46.8, 49.6]	48.5 (6.86) [46.9, 50.1]	46.7 (5.62) [43.6, 49.8]	0.199	0.365
Leg CG (cm)		32.5 (3.32) [31.8, 33.3]	32.8 (3.39) [32.0, 33.6]	31.4 (2.75) [29.9, 32.9]	0.282	0.281
∑3CG (mm)		107.0 (14.4) [104, 110]	108.0 (14.8) [105, 112]	104.0 (12.1) [97, 110]	0.185	0.463
∑3D (mm)		20.7 (1.51) [20.3, 21.0]	20.7 (1.45) [20.4, 21.1]	20.3 (1.78) [19.3, 21.3]	0.205	0.356
REE (kcal)		1796 (415) [1707, 1885]	1797 (412) [1700, 1895]	1791 (443) [1545, 2036]	0.112	0.821

Data are presented as mean (standard deviation) unless otherwise indicated. ξ: explanatory measure of effect size; ∑6S: sum of six skinfolds; ∑8S: sum of eight skinfolds; BM/W: body mass-to-waist ratio; CG: corrected girth; CI: confidence interval; EDG: equation development group; MoE_Δ_: margin of error for the CI on the difference between the two trimmed means; PAL: physical activity level; VG: validation group; W/Stature: waist-to-stature ratio. Statistical significance (*p* < 0.05 of the two-tailed *p* value) for the Yuen-Dixon test would indicate a difference between EDG and VG.

**Table 2 nutrients-16-03121-t002:** Regression results of the selected model using indirect calorimetry as the criterion.

Predictor	b [95% CI]	beta [95% CI]	*r*
(Intercept)	386.26 [−567.02, 1339.54]		
Age	−7.42 [−17.04, 2.21]	−0.14 [−0.31, 0.04]	0.12
BM	24.31 [11.74, 36.88] **	0.82 [0.40, 1.25]	0.68 **
Corrected Calf	38.63 [−1.11, 78.37]	0.32 [−0.01, 0.64]	0.67 **
Corrected Thigh	−21.35 [−44.71, 2.01]	−0.36 [−0.74, 0.03]	0.60 **
∑8S	−2.40 [−4.54, −0.26] *	−0.22 [−0.43, −0.02]	−0.12

A significant b-weight indicates the beta-weight correlation is also significant. *b*: represents unstandardized regression weights; beta: indicates the standardized regression weights; BM: body mass; ∑8S: sum of eight skinfolds. Statistical significance (* *p* < 0.05, ** *p* < 0.01).

## Data Availability

Data and statistical analyses are available for non-commercial scientific inquiry and/or educational purposes if requested and use does not violate IRB restrictions and/or research agreement terms.

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
