# Peer review of "Sum of Skinfold-Corrected Girths Correlates with Resting Energy Expenditure: Development of the NRG_CO_ Equation"

_nutrients, 2024, doi:10.3390/nu16183121_

Round 1
Reviewer 1 Report (Previous Reviewer 3)
Comments and Suggestions for Authors
My final recommendations are the following:
In the Participants section, I recommend mentioning the number of subjects included in the study and referencing the data about them to table 1.
2.5.3. Physical Activity Level - I recommend to mention the reliability of the applied questionnaire by calculating the α-Cronbach index, related to the answers of the study subjects.
Author Response
Dear reviewer,
Please find attached the point-by-point response to your comments/suggestions. Thanks for your feedback.
Sincerely,
The Authors

Reviewer 2 Report (Previous Reviewer 2)
Comments and Suggestions for Authors
Dear Sirs,
From my point of view, the changes made are correct.
Regards.
Author Response
Comments 1:
Dear Sirs,
From my point of view, the changes made are correct.
Regards.
Response 1:
Dear Reviewer,
Thank you for your positive feedback and for taking the time to review our manuscript. We are pleased to hear that the changes made have met your expectations.
Sincerely,
The authors
Reviewer 3 Report (Previous Reviewer 1)
Comments and Suggestions for Authors
Review of the resubmitted manuscript: nutrients-3139057
Sum of Skinfold-corrected Girths correlates with Resting Energy Expenditure: Development of the NRGCO Equation
The manuscript has been substantially modified/improved and in my opinion, the manuscript is ready for publication in the esteemed journal “Nutrients”.
Author Response
Comments 1:
Review of the resubmitted manuscript: nutrients-3139057
Sum of Skinfold-corrected Girths correlates with Resting Energy Expenditure: Development of the NRGCO Equation
The manuscript has been substantially modified/improved and in my opinion, the manuscript is ready for publication in the esteemed journal “Nutrients”.
Response 1:
Dear Reviewer,
Thank you for your positive feedback and for taking the time to review our manuscript. We are pleased to hear that the changes made have met your expectations.
Sincerely,
The Authors
This manuscript is a resubmission of an earlier submission. The following is a list of the peer review reports and author responses from that submission.
Round 1
Reviewer 1 Report
Comments and Suggestions for Authors
Review of the manuscript: nutrients-3139057
Sum of Skinfold-corrected Girths correlates with Resting Energy Expenditure: Development of the NRGCO Equation
The aim of the study was to “to compare methods for estimating REE in Colombian men and women with a moderate-to-high physical activity level” and “to conduct the external validation of the equations commonly used in clinical practice”. The Authors also wanted to develope “a novel equation called NRGCO“.
The Authors hypothesized that “the new NRGCO equation will allow for the estimation of REE based on simple anthropometric variables associated with musculoskeletal mass, sex, and/or basic measurements (body mass and stature).”
The Authors have demonstrated “that equations commonly used in clinical practice to estimate REE (Harris-Benedict, Mifflin-St. Jeor, and WHO) might not be valid for physically active Colombian men and women.”. They have developed a new equation, called NRGCO, to estimate resting metabolic rate in Colombian population with moderate-to-high physically activity levels.
The topic of the manuscript is very interesting. The study design, methods and results are well described. Statistical analysis is properly applied.
I have, however, some remarks to the manuscript.
1) Due to the small participant study group as well as its specific populations, I propose to add “Pilot Study” to the title of the paper.
2) What version (short or long) of IPAQ has been used?
3) Line 217 – should be Figure 1 not 2
4) Line 272 – put dot between Supplementary and Table
5) Line 446 and 447 – the citation should be corrected.
Reviewer 2 Report
Comments and Suggestions for Authors
Dear people, I hope to contribute to improving the work done.
The first part of the introduction seems correct to me (up to line 60).
I think that in the following lines it is necessary to incorporate more background information on the problems that existing formulas have. When you state that several equations have been developed, I think it is convenient to detail some of them (perhaps the most used) and give arguments why they would not adapt to the population for which you want to develop (something similar to what you have between lines 61 to 67, but in greater detail).
In line 68 they name the population of athletes. My doubt is that all the existing formulas are for people in general and also for athletes. Or the best thing is to occupy a specific one for athletes.
Depending on the answer to what I ask, they should determine how to proceed in terms of athletes. If its formula will still be for athletes, it is necessary to provide more background information on its characteristics, and the need to create something new for them.
The objective of their research is to compare "methods", which they name in the following lines. I think it is necessary to specify the attributes of the chosen methods.
2.5.3 state that the participants will be classified into three categories, where the sedentary ones are. Here I think there is something that needs to be improved, since the objective exposes something different. Check and correct.
The description of the results seems correct to me, as does the writing of the discussion chapter.
As a summary, I believe it is necessary to make major changes to the introduction, which will provide more background on the need to carry out the study.
Greetings.
Reviewer 3 Report
Comments and Suggestions for Authors
The interesting idea of this study, my recommendations are the following:
Abstract - I recommend mentioning what the values mentioned in line 27 represent - the arithmetic mean and the standard deviation. I recommend to mention descriptively what the acronyms ISAK and REE represent. I recommend to mention the values with two decimal places as in the study.
Lines 201-202 recommend moving to the Participants section.
According to table 1, the first 3 indicators mention the number and %, I recommend to mention the percentages in the top line of the table.
I recommend that the limits of this study and the practical implications be mentioned at the end of the Discussion section. I also recommend mentioning future research directions.
In conclusion, the study is clear, logical and coherently organized, the results are logically presented.